# Extended Gravity Constraints at Different Scales

**Stanislav Alexeyev** [1,2,*] **and Vyacheslav Prokopov** [1,3]

1 Sternberg Astronomical Institute, Lomonosov Moscow State University, Universitetskii Prospekt, 13, 119234 Moscow, Russia; slaprok777@gmail.com
2 Department of Quantum Theory and High Energy Physics, Physics Faculty, Lomonosov Moscow State University, Leninskie Gory, 1/2, 119234 Moscow, Russia
3 Department of Astrophysics and Stellar Astronomy, Physics Faculty, Lomonosov Moscow State University, Leninskie Gory, 1/2, 119234 Moscow, Russia
* Correspondence: alexeyev@sai.msu.ru
† This paper is an extended version of our paper published in An extended version of a conference paper The paper represents an extended version of the lecture presented by SA at XXII International Meeting "Physical Interpretations of Relativity Theory-2021" (5–9 July 2021), held at Bauman Moscow State Technical University.

**Simple Summary:** Simple summary We review a set of the possible ways to constrain extended gravity models at Galaxy clusters scales (the regime of dark energy explanations and comparison with ΛCDM), for black hole shadows, gravitational wave astronomy, binary pulsars, the Solar system and a Large Hadron Collider (consequences for high-energy physics at TeV scale).

**Abstract:** We review a set of the possible ways to constrain extended gravity models at Galaxy clusters scales (the regime of dark energy explanations and comparison with ΛCDM), for black hole shadows, gravitational wave astronomy, binary pulsars, the Solar system and a Large Hadron Collider (consequences for high-energy physics at TeV scale). The key idea is that modern experimental and observational precise data provide us with the chance to go beyond general relativity.

**Keywords:** general relativity; extended gravity; black hole; turnaround radius; shadow of black hole; gravitational waves; binary pulsars

**PACS:** 04.50.+h; 04.50.Gh; 04.80.Cc

## 1. Introduction

The theory of General Relativity (GR) is confirmed in all projects of experimental astronomy. However, the problems of dark energy, dark matter, the evolution of the early Universe, and the quantum theory of gravity remain open. For example, the theoretical description of the Universe's accelerated expansion (i.e., dark energy) is realised by adding the cosmological constant to the GR action $L$ as

$$L_{GR\Lambda} = \sqrt{-g}(R + \Lambda),\tag{1}$$

where $R$ is Ricci scalar and $\Lambda$ is the cosmological constant. The problem is that $\Lambda$-term is the best fit for the observational data. On the other hand, from the fundamental point of view, it appears to be a pure fine-tuning parameter. The next step is to consider an additional scalar field $\phi$ in the form of Brans–Dicke model

$$L_{BD} = \sqrt{-g}\left(\phi R + \frac{\omega}{\phi}\partial_\mu \phi \partial^\mu \phi + V(\phi)\right).\tag{2}$$

Such a model can reproduce the cosmological constant contribution with the help of taking the appropriate form of $V(\phi)$. Now, one has to find the origin of the scalar field in Equation (2). The same problem occurs with the inflation stage: accelerated expansion of

the early Universe. Mathematically, the power law asymptote of scale factor is changed to an exponential one. The inflation also can be modelled by the model, as in Equation (2) and in such a case, the scalar field is called "inflaton". As a consequence, one meets the question on the fundamental physical origin of inflaton. Of course, the list of such phenomena is more wide. Therefore, the extension of GR by additional physical fields or curvature terms [1] could be the next step in finding the origin of these phenomena.

GR extending could proceed in different ways. One can add the curvature invariants or pure degrees of scalar curvature and obtain $f(R)$ gravity [2]. These curvature corrections can reproduce the necessary behaviour, but once again, one meets the question of their origin. About ten years ago, the application of Horndesky theory [3,4] (the most general form of scalar tensor gravity with second-order field equations) became very popular. The successful development of Horndesky theory was corrected by GW170817 event [5]. So, a Binary Neutron Star Merger was detected. In addition to gravitational wave, an electromagnetic signal was also registered. Based on the time delay (about 2 s) between these types of radiation arrivals, the graviton mass value was limited [6,7]. This limitation made a cutoff of a big class of extended gravity models where graviton mass appeared to be greater. Therefore, during the last few years, a set of beyond Horndesky models forming a class of Degenerate Higher-Order Scalar–Tensor theories (DHOST) was developed [8]. The ideas of $f(R)$ gravity as more simple model [2] were also developed. The set of extended gravity models is wider. We restrict ourselves to the discussion on scalar–tensor gravity models, as they developed rather well, and the corresponding constraints look maximally clear. Of course, the list of models and corresponding constraints could be continued.

Note that currently, there is no preferred extended gravity model. A lot of versions in each class exist. A possible way to narrow down the amount of extended gravity models is to compare these predictions with real astrophysical data [9,10]. To extract the models that give more accurate predictions or require less fine-tuning, it is desirable to consider the maximally wide range of energies and distances: from galaxy clusters to high-energy physics. Therefore, we discuss a set of astrophysical tests for extended gravity models. Note that this list is far from being complete and represents only a small set of examples (open for extension).

We start the consideration from the scales of galaxy clusters. At these ranges, the contribution of accelerated expansion (dark energy) becomes considerable. Earlier, it was suggested to use the turn-around radius [11,12]. Recall that turnaround radius is a hypothetical surface where the internal gravitational force is equated by the accelerated expansion one. From one side, the value of the turnaround radius could be estimated using observational data on cluster sizes. Such assumptions could be obtained from gravitational lensing. From the other side, this value can be directly calculated from GR. As a first approximation, spherically symmetric space–time could be taken. Extended gravity models has different spherically symmetric metrics so one obtains the possibility of comparing the calculated values with observational ones to find the best correspondence.

The next step is to check the predictions of extended gravity models for shadows of black holes (BH). From one side, the Event Horizon Telescope is providing images for M87 [13] with increasing accuracy [14]. From the other side, the shadow size, form, and other characters depend on BH solution. As BH metrics are specific to each extended gravity model, here is the other possibility to compare the predictions of extended gravity models with Event Horizon Telescope observational results.

The same approach is applicable in order to find the constraints from gravitational wave astronomy [15]. The LIGO collaboration continues to collect data [16] on neutron stars and black-hole mergers [17]. From the other side, different extended gravity models provide different descriptions for gravitational waves, so their comparison could provide new information. Continuation of astronomical tests seems to be impossible without the most accurate data in astronomy: pulsar timing [18], especially for PSR-1913+16 [19]. The idea to use pulsar data for strong field tests of relativistic gravity [20] appeared to be very fruitful. Now it can be applied for extended gravity models. The idea is the same

as previously: to reproduce the values of post-Keplerian formalism using the specific extended gravity model parameters, then to extract the model that better reproduces the observational results [21].

The decreasing of the distances leads us to the Solar System ones. The parametric post-Newtonian (PPN) formalism [22] appears to be very effective to constrain different gravity models. In the Solar System, one has a competition between the maximal experimental accuracy versus vanishing additions to GR coming from extended gravity, especially while using the last data [23].

It is impossible not to mention the ideas of gravity constraining using high-energy physics data from the Large Hadron Collider (LHC). The well-known idea to search black holes at LHC [24] (and to discriminate some theories [25]) is not very popular now. Anyway, LHC data help to put some limitations on gravity in a quantum regime [26].

The paper is organised as follows. Section 2 is devoted to the difference between extended gravity predictions and general relativity ones at galaxy clusters scales (the regime of dark energy explanations and comparison with ΛCDM); Section 3 deals with black hole shadows; Section 4 is devoted to gravitational wave astronomy; Section 5 deals with binary pulsars; Section 6 is devoted to the Solar system; Section 7 deals with consequences from experimental high-energy physics at TeV scale and Section 8 includes the concluding remarks.

## 2. Galaxy Clusters Scales: Dark Energy Explanations

Nowadays the ΛCDM model with the action

$$S = \frac{1}{16\pi} \int d^4x \sqrt{-g}(R + \Lambda),$$
(3)

where $g_{\mu\nu}$ is the space–time metric (with determinant $g$), $R$ is the Ricci scalar, and $\Lambda$ is the cosmological constant, is the best fit for observational data, and provides a good description of dark energy. So to suggest a good physical interpretation of cosmological constant, it seems useful to modify GR [27]. Therefore, the models of $f(R)$ gravity (where $\Lambda$ is treated as a manifestation of knotty space–time geometry), scalar–tensor ones (where in addition to the previous one deals with additional fields), modern teleparallel ones (where the complex fundamental geometry is explored) and other ideas are developing. The first step in each GR extension leads to Brans–Dicke model

$$S = \frac{1}{16\pi} \int d^4x \sqrt{-g} \left( \phi R - \frac{\omega}{\phi} g^{\mu\nu} \nabla_\mu \phi \nabla_\nu \phi - V(\phi) + L_{matter} \right),$$
(4)

where $\phi$ is scalar field, $\omega$ is Brans–Dicke parameter, $V(\phi)$ is field potential and $L_{matter}$ is the contribution of matter fields. The model (4) reproduces the contribution of $\Lambda$. On the other hand the question on the origin of $\Lambda$ changes to the same one on $\phi$. That is why it is necessary to go further, for example, to $f(R)$ gravity [28]. In the frames of $f(R)$ gravity, a very interesting model was presented [29]. Being a good fit for $\Lambda$, this model is called a Starobinsky model with a disappearing cosmological constant. Note that further on, the continuation of this model appeared [30]. This approach allows us to provide the unified theory both for inflation and dark energy. So the Starobinsky model with disappearing cosmological constant is described by the following Lagrange density:

$$f(R) = R + \lambda R_0 \left( \left[ 1 + \frac{R^2}{R_0^2} \right]^{-n} - 1 \right),$$
(5)

where $R_0$ and $n$ are model parameters. Using the Starobinsky model with a disappearing cosmological constant as an example, we demonstrate how to constrain the theory at extra-galactic distances.

At the range of galaxy clusters, the contribution of Universe-accelerated expansion is considerable; therefore, the application of the turn-around radius [11,31] looks prospective. Recall that the turnaround radius is a hypothetical surface where the internal gravitational forces are equated by the accelerated expansion ones. The value of the turnaround radius is estimated from observational data, namely cluster size estimates using gravitational lensing (see, for example, [32–34]). On the other hand, the value of the turnaround radius can be calculated using spherically symmetric space–time as the first approximation. Different extended gravity models result in different versions of spherically symmetric metrics. Thus, it is possible to compare these calculated values with observational estimations to find the best correspondence. Note that there are a lot of activity in this field, for example, [35–44].

As an example, we show the procedure of the Starobinsky model with vanishing cosmological model check-up[12]. As a first approximation, it is convenient to take the metric in quasi-Schwarzschild form:

$$ds^2 = e^A dt^2 - e^{-A} dr^2 - r^2 d\Omega, \tag{6}$$

where $A = A(r)$ is metric function (for Schwarzschild case $A = 1 - 2M/r$). It is necessary to note that the usage of Schwarzschild metric (6) here is an approximation. Really, spherically symmetric solutions for the extended models must have two or more metric functions [45]. As the Schwarzschild metric could be treated as first terms of the Taylor expansion, correspondingly $r^{-1}$ the application of the Schwarzschild metric here could be treated as the first approximation. Formally, this analysis has to be extended. We restrict this procedure because of the observational data errors. So, the corresponding field equations are [46]:

$$f'(R)R_{ii} - f(R)\frac{g_{ii}}{2} - (\nabla_i^2 - g_{ii}\Box)f'(R) = 0. \tag{7}$$

At the turnaround radius, the first derivative of gravitational potential

$$\phi = \frac{1}{2}(g_{00} - 1) = \frac{1}{2}(e^A - 1) \tag{8}$$

must vanish:

$$\frac{dA}{dr} = 0. \tag{9}$$

Solving Equation (9) together with Equation (7) numerically, one finds the dependence of turnaround radius versus different values of $n$ from Equation (5). The most interesting mass range begins from $10^{11} M_{Sun}$ (Milky Way) and finishes at $10^{15} M_{Sun}$ (galaxy clusters). The results are demonstrated at Figure 1, from which one concludes that fewer values of $n$ provide better approximation of observational data. The same analysis could proceed for other extended gravity models; for example, the very rough approximation for Horndesky theory is presented in [47].

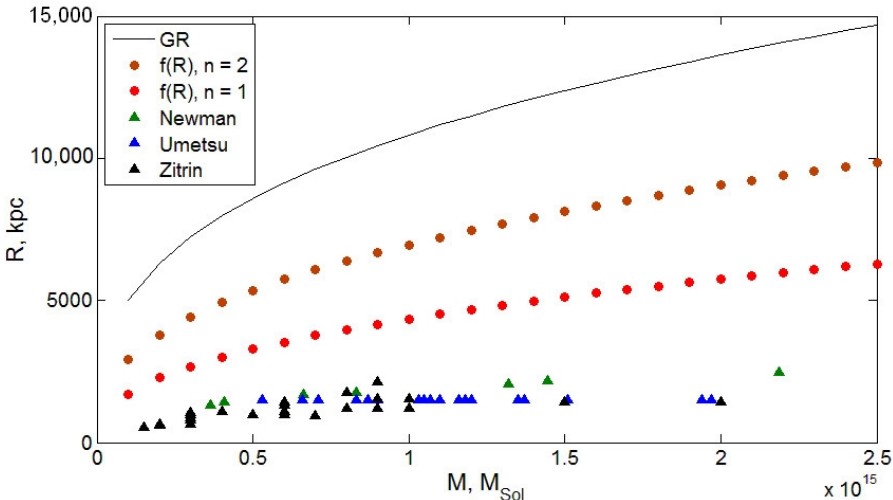

**Figure 1.** The theoretical and observational assumptions for the dependence of the turnaround radius value upon the mass of the galaxy cluster. The solid line corresponds to GR. Dots show these values for Starobinsky model with $n = 1$ and $n = 2$. Triangles show the real data from [32–34]. Reprinted/adapted with permission from Ref. [12]. 2017, J. Exp. Theor. Phys.

### 3. Black Hole Shadows: Deviations from GR

The idea to use the images of BH shadows for testing of extended gravity theories developed during the last 10 years [48,49]. Note that the same form of BH shadow can be obtained in the frames of different gravity theories. Usually, GR solutions (as the simplest ones) are taken as the first approximation. Note that there is a lot of activity in this field; for example [50–63]. So, a non-rotating, non-charged (Schwarzschild) BH (in the Planck system of units $G = c = \hbar = 1$) is described by the following metric:

$$ds^2 = -A(r)dt^2 + B(r)dr^2 + r^2(d\theta^2 + \sin^2\theta d\phi^2), \tag{10}$$

where

$$A(r) = B(r)^{-1} = 1 - \frac{2M}{r}, \tag{11}$$

$M$ is the mass of the BH. If the electromagnetic field (or new physics contribution) is taken into account, the Reissner–Nordstrom metric appears to be valid:

$$A(r) = B(r)^{-1} = 1 - \frac{2M}{r} + \frac{Q^2}{r^2}, \tag{12}$$

where $Q$ is the electric or tidal charge. The rotation can be included with the help of the Newman–Janis algorithm [64]. Applying it to (11) one results in the Kerr–Newman metric in the form:

$$ds^2 = -\left(1 - \frac{2m(r)r}{\rho^2}\right)dt^2 - \frac{4m(r)ar\sin^2\theta}{\rho^2}d\phi dt + \frac{\rho^2}{\Delta}dr^2 + \rho^2 d\theta^2$$
$$+ \left(r^2 + a^2 + \frac{2m(r)a^2 r\sin^2\theta}{\rho^2}\right)\sin^2\theta d\phi^2, \tag{13}$$

where

$$\rho^2 = r^2 + a^2\cos^2\theta,$$
$$\Delta(r) = r^2 - 2m(r)r + a^2,$$
$$m(r) = M - \frac{Q^2}{2r}, \tag{14}$$

$a = J/M$ is the BH acceleration and $J$ is its angular momentum. If $Q = 0$ one obtains an uncharged rotating BH described by the Kerr metric. For the Reissner–Nordstrom metric ($q \to Q^2$), the radius of BH shadow was calculated analytically [48] and is equal to

$$D = \sqrt{\frac{(8q^2 - 36q + 27) + \sqrt{(8q^2 - 36q + 27)^2 + 64q^3(1 - q)}}{2(1 - q)}}. \tag{15}$$

When $q > 1$, the object has no horizon and the minimum size of the BH shadow is equal to $4M$. From (15), one sees that for $q > 9/8$, a photon sphere is absent. For more complicated theories, it is impossible to obtain the analytical solution, so the numerical methods must be used.

For the space–time (10) in general $A(r) \neq -B(r)^{-1}$. Corrections from extended theories can be represented as additional terms of the Taylor series expansion around the Schwarzschild metric. Therefore, to simulate the shadow, one has to study the equations of motion of the photons around the BH. So,

$$u(r) = \left(\frac{d\hat{r}}{d\phi}\right)^2 = \frac{\hat{r}^4}{D^2 A(\hat{r}) B(\hat{r})} - \frac{\hat{r}^2}{B(\hat{r})}, \tag{16}$$

where $D = L/E$ is the aiming parameter of the photon beam. The edge of the shadow corresponds to the transition of light particles to an unstable orbit. The conditions of this transition are:

$$u(r) = 0, \quad \frac{du(r)}{dr} = 0, \quad \frac{d^2 u(r)}{d^2 r} > 0 \tag{17}$$

The size of the shadow is defined by the the maximal solution of Equations (17). The accounting of $1/r^{-3}$ correction allows to cover the wider range of shadow sizes. For example, it becomes possible to study BH shadow with a size less than $4M$ [65,66]. Moreover, the objects with a horizon but without the photon sphere become well described. This is a BH without a shadow appearing in the models with beyond Reissner–Nordstrom metrics.

It is important to note that many theories predict the existence of BH shadows with the same size. To determine the theory type, additional tests of the BH potential are required. They are, for example, strong gravitational lensing of bright objects around the BH, the last stable orbit of the accretion disk, the distribution of background intensity from ionised plasma around the BH, etc. Earlier [65,66] we estimated an accuracy value that is enough to constrain extended theory in a BH non-rotating case. For strong gravitational lensing, this accuracy is about $10^3$ times less than the angular size of the BH required. For the background intensity, the necessary accuracy is also $\approx 10^3$ times less than the its maximum value.

One of the ways to include rotation is to apply the Newman–Janis algorithm [67]. The usage of $m(r)$ term allows us to incorporate various models. Generically, for the corrections established as Taylor series, the procedure is as follows:

$$m(r) = M - \frac{q}{2r} - \frac{C_3}{2r^2} - \ldots - \frac{C_n}{2r^{n-1}} - \ldots. \tag{18}$$

Therefore, the coordinates of the shadow edge $[\alpha, \beta]$ on the image plane are:

$$\alpha = \frac{\xi_-}{\sin \theta_i},$$

$$\beta = \pm \sqrt{\eta_- + (a - \xi_-)^2 - \left(a \sin \theta_i - \frac{\xi_-}{\sin \theta_i}\right)^2}, \tag{19}$$

where $\theta_i$ is the angle of inclination of the BH rotation axis,

$$\xi_- = \frac{4r_0^2\xi_A - (r_0^2 + a^2)\xi_B}{a\xi_C},$$

$$\eta_- = \frac{r_0^3[\eta_A a^2 - r_0\eta_B^2]}{a^2\eta_C^2}, \tag{20}$$

$$\xi_A = M - \frac{q}{2r_0} - \frac{C_3}{2r_0^2} - \ldots - \frac{C_n}{2r_0^{n-1}} - \ldots,$$

$$\xi_B = r_0 + M + \frac{C_3}{2r_0^2} + \ldots + \frac{(n-2)C_n}{2r_0^{n-1}} + \ldots,$$

$$\xi_C = r_0 - M - \frac{C_3}{2r_0^2} - \ldots - \frac{(n-2)C_n}{2r_0^{n-1}} - \ldots,$$

$$\eta_A = 4M - \frac{4q}{r_0} - \frac{6C_3}{r_0^2} - \ldots - \frac{2nC_n}{r_0^{n-1}} - \ldots,$$

$$\eta_B = r_0 - 3M + \frac{2q}{r_0} + \frac{5C_3}{2r_0^2} + \ldots$$
$$+ \frac{(n+2)C_n}{2r_0^{n-1}} + \ldots,$$

$$\eta_C = r_0 - M - \frac{C_3}{2r_0^2} - \ldots - \frac{(n-2)C_n}{2r_0^{n-1}} - \ldots,$$

For the corrections in the form of Taylor expansion, the accuracy was estimated earlier [68] where the shapes of BH shadows with the same size have been compared. The deviation of the shape from the Kerr BH one reaches 2% of the shadow size (Figure 2) for the expansions up to $r^{-3}$, strong rotation with $a = 0.9$ and for $q$ and $C_3$ comparable with $M$ (for example, $q = 0.17$, $C_3 = -0.5$). To constrain rapidly rotating BH, the accuracy an order of magnitude better is required. This is about one hundred times louder than the size of the shadow.

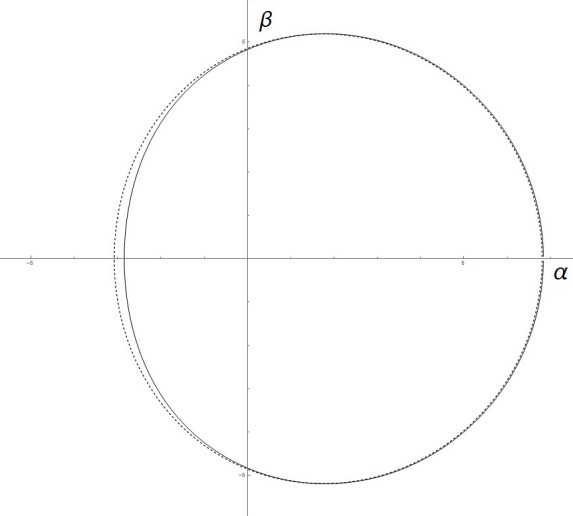

**Figure 2.** Black holes with the spin equal to $a = 0.9$ and the following parameters: dashed line corresponds to $q = 0.17$, $C_3 = -0.5$; solid one corresponds to $q = 0$, $C_3 = 0$. The rotation axis is directed along the $\beta$ one; the inclination angle is equal to $\theta_i = -\pi/2$. Reprinted/adapted with permission from Ref. [68]. Copyright year: 2020, copyright owner's name: J. Exp. Theor. Phys.

As a result: in order to test gravity theories, one needs the resolution to be hundreds of times better than was reached in[13]. The current radio telescopes are already used in one big web. So, the next step to improve the resolution is to use space telescopes.

## 4. Gravitational Wave Astronomy: Deviations from GR

One of the most well-known results in experimental astronomy is the registration of gravitational waves [69]. From a mathematical point of view, the gravitational wave is a solution of GR and the existing experimental results completely correspond to it. The LIGO collaboration continues to collect data [16] on neutron stars and black holes mergers [17]. The most interesting event for our aims was GW170817 [70], when two neutron stars merged not far from us. Thanks to this, the direction to the source was discovered, so the corresponding electromagnetic signal was also registered [71]. The time delay between gravitational and electromagnetic signals was equal to 1.6 s. Such a time-delay value puts a strong limitation on graviton mass [6]. Therefore, it makes a great cutoff for the models of massive gravity. Based on these data, graviton mass appears to be less than $10^{-22}$ eV. Hence, the difference between the speeds of light and gravitational wave could not be more than $1 + 10^{-16}$. Such a restriction excludes a big set of models with massive graviton (and, therefore, models where it could appear). After GW170817, such models as quartic/quintic galileons, models Fab Four (for additional clarifications see [72]), some Degenerated Higher-Order Scalar Tensor (DHOST) models with $A_1 \neq 0$, and many others appeared to be excluded [6,7]. Horndesky models (as general version of scalar–tensor gravity with second order field equations [3,4]), which for a long time were the top candidate for dark energy and dark matter explanations after GW170817, are restricted as $G_{4,X} = 0$, $G_5 = const$ [6]. Therefore, to follow the GW170817 limit, a new set of models beyond Horndesky was developed. There are such theories as derivative conformal models, models with disformal tuning, and DHOST models with with $A_1 = 0$, and so on [6,7]. Of course, simpler theories such as Brans–Dicke and $f(R)$ models remain in use. Finally, each viable gravitational model must pass the gravitational wave astronomy tests (including LIGO-VIRGO last runs).

Note that the subject of how gravitational wave astronomy could constrain extended gravity models is much wider. We restrict our consideration with the GW170817 test, as it seems to be the most intriguing. One can extend the discussion, for example, by the notes on the forms of gravitational wave solutions in different models and the number of polarisation modes.

## 5. Binary Pulsars: Deviations from GR

Any discussion on astronomical tests would be incomplete without mentioning the most accurate astrophysical data set: pulsar timing [18]. The most accurate data are provided by PSR-1913+16 [19]. The key idea remains the same: to reproduce all post-Keplerian parameters using the specific extended gravity model. Then one has to find the model which covers the observational results better[21]. Note that in binary pulsars, one deals with a gravitational field much stronger than the one in the Solar system. It lies closer to the one discussed in the previous section. Thanks to the stability of the pulse signal, one can extract the orbital motion dynamics and gravitational waves emission contributions. There is a lot of activity in this field; for example [6,9,73,74].

Following [20,75,76], we start from the famous timing formula:

$$t_B - t_0 = D^{-1}\left[T + \Delta_R(T, \dot{\omega}, \dot{P}_b, \delta_r, \delta_\theta) + \Delta_E(T, \gamma) + \Delta_S(T, r, s) + \Delta_A(T, A, B)\right], \quad (21)$$

where $t_B$ is the time of arrival of an impulse at the barycenter of the solar system, $t_0$ is the observable time of impulse arrival, $D$ is the Doppler factor, $T$ is the time of impulse emission, and ($\Delta_R$, $\Delta_E$, $\Delta_S$, and $\Delta_A$) are the propagation delays due to "Roemer", "Einstein", "Shapiro" (see, in addition, [77]) and "aberattion" effects, respectively. These delays depend upon the parametrised Keplerian and post-Keplerian (PPK) parameters ($\omega$, $P_b$, $\delta_r$, $\delta_\theta$, $\gamma$,

*r*, *s*, *A*, *B*) are periastron longitude, pulsar orbital period and its first derivative, two parameters of orbit deformation, Einstein delay, two parameters of Shapiro delay, two aberration parameters. The strategy of these values calculation is presented in lots of papers following the original Damour and Taylor ones. It is important to emphasise that these PPK parameters are calculated using metric decomposition and power series. Hence, the result for each model is unique. The ideas to constrain extended gravity models were developed [78], and it was shown that GR ideally passes through the ($\dot{P}_b - \dot{\omega} - \gamma$) test if the existence of gravitational waves is taken into account.

Let us concentrate on $\dot{P}_B$ version. In [21] its form was calculated for the general version of scalar–tensor gravity, i.e., the Horndesky model. So, the $\dot{P}_b^{th}/\dot{P}_B^{GR}$ relation where $\dot{P}_b^{th}$ is the first derivative of the orbital period for the theory under consideration and $P_B^{\dot{G}R}$ is the same value for GR can be established as:

$$\frac{\dot{P}_b^{th}}{\dot{P}_b^{GR}} = \frac{\mathcal{G}_{12}^{\frac{2}{3}}}{G^{\frac{5}{3}}G_{4(0,0)}}\left\{1 + \frac{5G_{4(1,0)}c_\varphi}{48}\left(\frac{P_b c^3}{2\pi m \mathcal{G}_{12}}\right)^{\frac{2}{3}} \times \left[A_d^2 + \frac{2\mu}{c^2}A_d \bar{A}_d\left(\frac{4\pi^2}{P_b^2 m \mathcal{G}_{12}}\right)^{\frac{1}{3}}\right]\left(1 - \frac{m_\varphi^2 c^2 P_b^2}{4\pi^2}\right)^{\frac{3}{2}}\right.$$
$$+ \frac{G_{4(1,0)}c_\varphi}{3}A_q^2\left(1 - \frac{m_\varphi^2 c^2 P_b^2}{16\pi^2}\right)^{\frac{5}{2}} - \frac{G_{4(1,0)}c_\varphi}{96}A_d A_o\left(1 - \frac{m_\varphi^2 c^2 P_b^2}{4\pi^2}\right)^{\frac{5}{2}}\right\}, \qquad (22)$$

where $\dot{P}_b^{GR}$ is:

$$\dot{P}_b^{GR} = -\frac{192\pi\mu}{5c^5 m}\left(\frac{2\pi Gm}{P_b}\right)^{\frac{5}{3}}, \qquad (23)$$

$G(i,j)$ are part of the Horndesky model [3,4] and $P_b$ is the standard expression for the orbital period. Based on Equation (22) and using the real data from $PSRJ1738 + 0333$, one can calculate constraints on different extended gravity models. So, the key formula allowing to calculate the limitations on the considered model is

$$\left|\frac{\dot{P}_b^{th}}{\dot{P}_b^{GR}} - \frac{\dot{P}_b^{obs}}{\dot{P}_b^{GR}}\right| \leq 2\sigma, \qquad (24)$$

where $\sigma$ is the observational uncertainty and $\dot{P}_b^{obs}/\dot{P}_b^{GR}$ is the observational quantity at 95% confidence level.

The first example is massive scalar–tensor gravity. Substituting its specific values and PSR J1738+0333 data to Equation (24), one obtains that

$$\left|\frac{\mathcal{G}_{12}^{\frac{2}{3}}}{G^{\frac{5}{3}}G_{4(0,0)}}\left[1 + \frac{5c_\varphi}{12}\left(\frac{P_b c^3}{2\pi m \mathcal{G}_{12}}\right)^{\frac{2}{3}}\left(1 - \frac{m_\varphi^2 c^2 P_b^2}{4\pi^2}\right)^{\frac{3}{2}} \times \left(\frac{G_{4(0,0)}^2(s_{NS} - s_{WD})^2}{G_{4(1,0)}\phi_0^2}\right)\right] - 0.93\right| \leq 0.26. \qquad (25)$$

Here, $s_{NS}$ is the sensitivity of a neutron star and $s_{WD}$ is the sensitivity of a white dwarf. So, one obtains the upper boundary for the scalar field mass $m_\varphi$:

$$m_\varphi < 7 \times 10^{-15}(\text{cm}^{-1}). \qquad (26)$$

The second example is hybrid metric-Palatini f(R) gravity. This model is developed as a mixture of metric and Palatini formalisms. The aim is to use the approach that provides the best theoretical description in the considered range. For example, the hybrid f(R)-gravity describes the accelerated Universe expansion without introducing of new degrees of freedom. Using the set of transfer parameters [21] one obtains that

$$0.67 \leq \frac{1}{(1 + \phi_0)^{\frac{5}{3}}}\left(1 - \frac{5\phi_0}{18}(1 - 3 \times 10^{27}m_\varphi^2)\right) \leq 1, \qquad (27)$$

where the dependence of the scalar field mass upon scalar field background value for PSR J1738+0333 can be taken from [21]. So, the combined restrictions from $\gamma_{PPN}$ and system PSR J1738+0333 are:

$$\phi_0 \leq 0.00004, \quad m_\varphi \leq 1.4 \times 10^{-14} (\text{cm}^{-1}). \tag{28}$$

The consideration could be extended [79] to constrain other theories.

## 6. Solar System: Newtonian Limit and Deviations from it

The Solar System contains a set of small parameters. They are, for example, Newtonian potential, matter velocity relative to the mass centre, etc. So, these values could be used as expansion parameters to consider the Taylor expansion of the metric. The standardised expansion represent PPN with the coefficients measured experimentally. We shall not discuss PPN formalism in detail, as there are a lot of beautiful reviews (see, for example, Ref. [80]) and textbooks ([22]). Here, we show the usage of PPN to constrain the hybrid metric-Palatini gravity.

The hybrid metric-Palatini f(R)-gravity is a part of f(R)-theories [2,81]. Indeed, there are two ways to obtain field equations: the metric one and the Palatini one. In the metric approach, the metric $g_{\mu\nu}$ is treated as the unique dynamical variable. The Palatini method supposes the Riemann curvature tensor is independent of the metric and dependent only upon connection. So, variations with respect to the metric and the connection become independent. Both gravity models, i.e., metric one and Palatini ones, cause problems. The metric f(R)-gravity in the general case does not pass the standard Solar System tests [82–84]. Palatini f(R) models contain microscopic matter instabilities [85,86]. In order to cancel these pathologies in both metric and Palatini formulations, the hybrid metric-Palatini f(R)-gravity was developed [87,88]. This model combines the GR Lagrangian and the $f(\Re)$-term constructed by the Palatini formalism. Therefore, all results obtained in the frames of GR are covered by the $R$ part and $f(\Re)$ part is responsible for unexplained gravitational phenomena.

Further, the hybrid f(R)-gravity can be established as a scalar–tensor theory [87,88]. If the appearing scalar field is light enough, it could modify the cosmological and galactic dynamics to cover unexplained phenomena, leaving the Solar System unaffected. So, it appears to be possible to provide a test of the hybrid f(R)-theory in the weak-field limit using the modified PPN formalism.

We start from the action [23,87,88]

$$S = \frac{c^4}{2k^2} \int d^4x \sqrt{-g}[R + f(\Re)] + S_m, \tag{29}$$

where $c$ is the speed of light, $k^2 = 8\pi G$, $R$ and $\Re = g^{\mu\nu}\Re_{\mu\nu}$ are the metric and Palatini curvatures, respectively, $g$ is the metric determinant, and $S_m$ is the matter action. The Palatini curvature $\Re$ is considered depending upon $g_{\mu\nu}$ and the independent connection $\hat{\Gamma}^\alpha_{\mu\nu}$:

$$\Re = g^{\mu\nu}\Re_{\mu\nu} = g^{\mu\nu}\left(\hat{\Gamma}^\alpha_{\mu\nu,\alpha} - \hat{\Gamma}^\alpha_{\mu\alpha,\nu} + \hat{\Gamma}^\alpha_{\alpha\lambda}\hat{\Gamma}^\lambda_{\mu\nu} - \hat{\Gamma}^\alpha_{\mu\lambda}\hat{\Gamma}^\lambda_{\alpha\nu}\right). \tag{30}$$

The discussed model allows the scalar–tensor representation in the Jordan frame in the form

$$S = \frac{c^4}{2k^2} \int d^4x \sqrt{-g}\left[(1+\phi)R + \frac{3}{2\phi}\partial_\mu\phi\partial^\mu\phi - V(\phi)\right] + S_m, \tag{31}$$

where $\phi$ is a scalar field and $V(\phi)$ is its potential. The following field equations are:

$$(1+\phi)R_{\mu\nu} = \frac{k^2}{c^4}\left(T_{\mu\nu} - \frac{1}{2}g_{\mu\nu}T\right) + \frac{1}{2}g_{\mu\nu}\left[V(\phi) + \nabla_\alpha\nabla^\alpha\phi\right] + \nabla_\mu\nabla_\nu\phi - \frac{3}{2\phi}\partial_\mu\phi\partial_\nu\phi, \tag{32}$$

$$\nabla_\mu\nabla^\mu\phi - \frac{1}{2\phi}\partial_\mu\phi\partial^\mu\phi - \frac{\phi[2V(\phi) - (1+\phi)V_\phi]}{3} = -\frac{k^2}{3c^4}\phi T. \tag{33}$$

Here, it is necessary to note that the scalar field is dynamical in the hybrid f(R)-gravity. So, there are no microscopic instabilities associated with in pure Palatini models.

We start by expanding of the scalar $\phi$ and the tensor $g_{\mu\nu}$ fields as

$$\phi = \phi_0 + \varphi, \qquad g_{\mu\nu} = \eta_{\mu\nu} + h_{\mu\nu}, \tag{34}$$

where $\phi_0$ is the field asymptotic value, $\eta_{\mu\nu}$ is the Minkowski space–time, and $h_{\mu\nu}$ and $\varphi$ are the small perturbations of tensor and scalar fields, respectively. In general, $\phi_0$ depends upon time. This dependence can be neglected if one deals with a short period associated with the observational time in comparison with the cosmological time-scale. So, we treat $\phi_0$ as a constant.

The PPN operates with the following orders of metric and field [22]:

$$\begin{aligned} h_{00} &\sim O(2) + O(4), \\ h_{0j} &\sim O(3), \\ h_{ij} &\sim O(2) \\ \varphi &\sim O(2) + O(4). \end{aligned}$$

Further, the Taylor expansion for the scalar potential $V(\phi)$ around the background value $\phi_0$ has the form:

$$V(\phi) = V_0 + V'\varphi + \frac{V''\varphi^2}{2!} + \frac{V'''\varphi^3}{3!}. \tag{35}$$

The stress–energy tensor for point-mass gravitational systems is defined as

$$T^{\mu\nu} = \frac{c}{\sqrt{-g}}\sum_a m_a \frac{u^\mu u^\nu}{u^0}\delta^3(\vec{r} - \vec{r}_a), \tag{36}$$

where $m_a$ is the mass of the $a$-th particle, $\vec{r}_a$ is its radius vector, $u^\mu = dx_a^\mu/d\tau_a$ is its four velocity, $d\tau = \sqrt{-ds^2}/c$, $ds^2 = g_{\mu\nu}dx^\mu dx^\nu$ is an interval, $u_\mu u^\mu = -c^2$, and $\delta^3(\vec{r} - \vec{r}_a(t))$ is the 3D Dirac delta function. In the PPN approximation, these components (36) and the trace $T$ have the following form:

$$T_{00} = c^2\sum_a m_a\delta^3(\vec{r} - \vec{r}_a)\left[1 - \frac{3}{2}h_{00} + \frac{1}{2}\frac{v_a^2}{c^2} - \frac{1}{2}h\right], \tag{37}$$

$$T_{0i} = -c\sum_a m_a v_a^i\delta^3(\vec{r} - \vec{r}_a), \tag{38}$$

$$T_{ij} = \sum_a m_a v_a^i v_a^j\delta^3(\vec{r} - \vec{r}_a), \tag{39}$$

$$T = -c^2\sum_a m_a\delta^3(\vec{r} - \vec{r}_a)\left[1 - \frac{1}{2}h_{00} - \frac{1}{2}\frac{v_a^2}{c^2} - \frac{1}{2}h\right], \tag{40}$$

where $v_a$ is the velocity of the $a$-th particle. At the next step, one uses the Nutku gauge conditions [89]:

$$h^\alpha_{\beta,\alpha} - \frac{1}{2}\delta^\alpha_\beta h^\mu_{\mu,\alpha} = \frac{\varphi_{,\beta}}{1+\phi_0}. \tag{41}$$

After solving the correspondent field equations [23] and assuming that the main contribution comes from the Sun, (here we restrict ourselves to the second-order expressions because of their length) one obtains the solution for $h_{00}^{(2)}$:

$$h_{00}^{(2)} = \frac{k^2}{4\pi(1+\phi_0)c^2} \frac{M}{r} \left(1 - \frac{\phi_0}{3} \exp[-m_\varphi r]\right) + \frac{V_0}{1+\phi_0} \frac{r^2}{6},$$

(42)

where $M$ is the Solar mass. Here, $V_0/(\phi_0 + 1)$ represents the $\Lambda$ term negligible in Solar System scales. In the same way,

$$h_{ij}^{(2)} = \frac{\delta_{ij}k^2}{4\pi(1+\phi_0)c^2} \frac{M}{r} \left(1 + \frac{\phi_0}{3} \exp[-m_\varphi r]\right) - \delta_{ij}\frac{V_0}{1+\phi_0} \frac{r^2}{6}.$$

(43)

After comparing the result expressions with the general point-mass form introduced by K. Nordtvedt, the expression for effective PPN parameter $\gamma^{\text{eff}}$ can be expressed as

$$\gamma^{\text{eff}} = \frac{1 + \phi_0 \exp[-m_\varphi r]/3}{1 - \phi_0 \exp[-m_\varphi r]/3}.$$

(44)

Taking the Equation (44) in a case of a light scalar field $m_\varphi r \ll 1$ and using the experimental values of PPN parameters one constrains $\phi_0$ as

$$-8 \times 10^{-5} < \phi_0 < 7 \times 10^{-5}$$

(45)

from the $\gamma^{\text{exp}}$ at the $2\sigma$ confidence level.

## 7. Large Hadron Collider: Constraints at TeV Scale

The last possibility to constrain extended gravity models that we discuss is the usage of high-energy physics data. Following [26], we intend to demonstrate how one could constrain an extended gravity model at the quantum gravity regime using LHC data.

We currently have no experimental data on matter properties at Planckian scale. From the theoretical considerations, it seems that the combination of Quantum Mechanics and GR may lead to a more complicated structure of space—time at short distances, so a new fundamental value as minimal length appears. Following the logic of Quantum Mechanic with uncertainty relation, one concludes that it is impossible to measure distances with a precision better than the Planck length $l_P = \sqrt{\hbar G/c^3}$ where $\hbar$ is the Planck constant, $G$ is the gravitational constant and $c$ is the speed of light in a vacuum. Using the LHC data, it appears possible to show that the scale of non-locality could actually be much larger than $l_P$.

Earlier, it was shown that GR coupled to a quantum field theory causes non-local effects in scalar field theories [90]. Here, we explore such a model with matter, including spinor and vector fields. Firstly, it is necessary to calculate a complete set of non-local effective operators at order $NG^2$, where $N = N_s + 3N_f + 12N_V$, $N_s$, $N_f$, and $N_V$ denote the number of scalar, spinor, and vector fields, respectively. Afterwards, one could obtain the possibility of constraining the scale of space–time non-locality with the help of recent data from the LHC. So, following [26], we start from perturbative linearised GR coupled to matter fields. Note that perturbative unitarity can be broken below the reduced Planck mass, but it can be recovered by the accurate resummation of a series of graviton vacuum polarisation diagrams in the large $N$ limit. The key feature of this large $N$ resummation is the following: if one could keep $NG$ small, therefore, the obtained graviton propagator ($\mu$ is the renormalisation scale)

$$iD^{\alpha\beta,\mu\nu}(q^2) = \frac{i\left(L^{\alpha\mu}L^{\beta\nu} + L^{\alpha\nu}L^{\beta\mu} - L^{\alpha\beta}L^{\mu\nu}\right)}{2q^2\left(1 - \frac{NGq^2}{120\pi}\log\left(-\frac{q^2}{\mu^2}\right)\right)},$$

(46)

includes some of the non-perturbative effects of quantum gravity. There are additional poles beyond the usual one at $q^2 = 0$. These complex poles are a sign of strong interactions and the mass and width of these objects can be calculated. These poles being complex could have an incorrect sign between mass and width. Therefore, this pole could be associated with a particle propagating backwards in time, leading to violation of causality. With the help of the in–in formalism [91,92] the causality restores. As a consequence, non-local effects at the scale $(120\pi/GN)^{1/2}$ arise, and so the scale of non-locality grows. It becomes larger than $l_P$ if there are many fields in the matter sector ($N$ is large).

Earlier [90], it was demonstrated that the resummation of a graviton propagator in Equation (46) leads to non-local effects in scalar field theories at the range of $(120\pi/GN)^{1/2}$. Such consideration should be extended to spinor and vector fields. Considering a model with an arbitrary number of scalar, spinor, and vector fields one has to calculate their two-by-two scattering gravitational amplitudes, taking into account the dressed graviton propagator (46). Next, the leading order ($G^2 N$) term should be extracted to present the results in terms of effective operators.

The stress–energy tensors for the different field species with spins 0, 1/2, and 1 are taken in the form [26]

$$T_{\text{scalar}}^{\mu\nu} = \partial^\mu \phi \, \partial^\nu \phi - \eta^{\mu\nu} L_{\text{scalar}} \, , \tag{47}$$

$$T_{\text{fermion}}^{\mu\nu} = \frac{i}{4}\bar{\psi}\gamma^\mu \nabla^\nu \psi + \frac{i}{4}\bar{\psi}\gamma^\nu \nabla^\mu \psi - \frac{i}{4}\nabla^\mu \bar{\psi}\gamma^\nu \psi - \frac{i}{4}\nabla^\nu \bar{\psi}\gamma^\mu \psi - \eta^{\mu\nu} L_{\text{fermion}} \, , \tag{48}$$

$$T_{\text{vector}}^{\mu\nu} = -F^{\mu\sigma} F^\nu_{\ \sigma} + m^2 A^\mu A^\nu - \eta^{\mu\nu} L_{\text{vector}} \, , \tag{49}$$

where the free field matter Lagrangians are:

$$L_{\text{scalar}} = \frac{1}{2}(\partial\phi)^2 - \frac{1}{2}m^2\phi^2 \, , \tag{50}$$

$$L_{\text{fermion}} = \frac{i}{2}\bar{\psi}\gamma^\sigma \nabla_\sigma \psi - \frac{i}{2}\nabla_\sigma \bar{\psi}\gamma^\sigma \psi - m\bar{\psi}\psi \, , \tag{51}$$

$$L_{\text{vector}} = -\frac{1}{4}F^2 + \frac{1}{2}m^2 A^2 \, . \tag{52}$$

The non-local operators at order $NG^2$ with scalar field look like (because of their length we present only one operator in each group, the other ones have the same structure, the complete set can be taken from [26]):

$$O_{\text{scalar},1} = \frac{NG^2}{30\pi}\partial_\mu\phi\partial_\nu\phi\ln\left(\frac{\Box}{\mu^2}\right)\partial^\mu\phi'\partial^\nu\phi',$$
$$\dots$$

The non-local operators with spinor fields are

$$O_{\text{fermion},1} = \frac{NG^2}{60\pi}\left(\frac{i}{2}\bar{\psi}\gamma^\mu \nabla^\nu \psi - \frac{i}{2}\nabla^\mu \bar{\psi}\gamma^\nu \psi\right)\left(\delta^\alpha_\mu \delta^\beta_\nu + \delta^\beta_\mu \delta^\alpha_\nu\right)\ln\left(\frac{\Box}{\mu^2}\right)\left(\frac{i}{2}\bar{\psi}'\gamma^\alpha \nabla^\beta \psi' - \frac{i}{2}\nabla^\alpha \bar{\psi}'\gamma^\beta \psi'\right),$$
$$\dots$$

The non-local operators involving vector fields only are:

$$O_{\text{vector},1} = \frac{NG^2}{30\pi}\left(F^{\mu\sigma} F_{\nu\sigma} - m^2 A^\mu A_\nu\right)\ln\left(\frac{\Box}{\mu^2}\right)\left(F'^{\mu\rho} F'_{\nu\rho} - m'^2 A'_\mu A'^\nu\right),$$
$$\dots$$

The non-local operators involving amplitudes with scalar and vector fields only are given by

$$O_{\text{scalar-vector},1} = -\frac{NG^2}{30\pi}\partial^\mu\phi\partial_\nu\phi\ln\left(\frac{\Box}{\mu^2}\right)\left(F_{\mu\sigma}F^{\nu\sigma} - m_A^2 A_\mu A^\nu\right),$$
$$\cdots$$

The non-local operators involving amplitudes with scalar and spinor fields only are:

$$O_{\text{scalar-fermion},1} = \frac{NG^2}{30\pi}\partial_\mu\phi\partial_\nu\phi\ln\left(\frac{\Box}{\mu^2}\right)\left(\frac{i}{2}\bar\psi\gamma^\mu\nabla^\nu\psi - \frac{i}{2}\nabla^\mu\bar\psi\gamma^\nu\psi\right),$$
$$\cdots$$

The non-local operators involving amplitudes with spinor and vector fields only are:

$$O_{\text{vector-fermion},1} = -\frac{NG^2}{30\pi}\left(\frac{i}{2}\bar\psi\gamma^\mu\nabla^\nu\psi - \frac{i}{2}\nabla^\mu\bar\psi\gamma^\nu\psi\right)\ln\left(\frac{\Box}{\mu^2}\right)\left(F_{\mu\sigma}F_\nu{}^\sigma - m_A^2 A_\mu A_\sigma\right),$$
$$\cdots$$

The given effective operators contain a non-local part in the form of $\ln(\Box/\mu^2)$ term in the matter sector. Its extension corresponds to the minimal length that can be tested. So, the space–time is smeared on distances shorter than $M_\star = M_P\sqrt{120\pi/N}$, which corresponds to the energy of the complex pole. Therefore, there is no correct definition of the space–time on distances smaller than $1/M_\star$. Note that the non-local effects in the four–fermion interactions can be constrained using LHC data. The ATLAS collaboration looked for four–fermion contact interactions at $\sqrt{s} = 8$ TeV and obtained lower limits on the scale on the lepton–lepton–quark–quark contact interaction $\Lambda$ between 15.4 TeV and 26.3 TeV [93]. The conservative approach suggests identifying the scale generated with the derivatives in the four–fermion operators with the centre of mass energy of the proton–proton collision. Therefore, the conservative bound could be taken as $N < 5 \times 10^{61}$ on the number of light fields in a hidden sector. As a result, the scale $M_\star$ (a character value of space–time non-locality) appears to be larger than $3 \times 10^{-11}$ GeV.

## 8. Conclusions

We briefly review the set of possibilities to constrain extended gravity models at different space–time and energy scales. The key idea is to apply modern astrophysical and high-energy physics, data starting from big scales, and to go down along the scale, paying attention to common features of all mentioned items. It is important to note that we briefly discuss a set in a unified way; to help one to carefully study each item, references to corresponding reviews are provided.

The first possibility occurs at the scales of galaxy clusters where the contribution of accelerated expansion (dark energy) becomes considerable. We focused on the turn-around radius usage. Extended gravity models have different spherically symmetric metrics, so one obtains the possibility of comparing the calculated values with observational ones to find the best correspondence. We showed that for the Starobinsky model with the vanishing cosmological constant, lower values of parameter $n$ provide better approximation for the observational data. For the Horndesky model, the upper limit on the effective cosmological constant value is $\Lambda_{eff} < 1.6 \cdot 10^{-48}$ m$^{-2}$ [47].

The next possibility to check the predictions of extended gravity models comes from shadows of black holes (BH). As we show, in order to test gravity theories, one needs to increase the resolution two orders of magnitude greater than was reached in [13].

The next possibility comes from gravitational wave astronomy. We mentioned that Horndesky models after GW170817 were restricted as $G_{4,X} = 0$, $G_5 = const$ so beyond Horndesky models began actively developing. Note that such theories as derivative

conformal models, models with disformal tuning and DHOST models with with $A_1 = 0$ remain valid.

The next possibility is given by the most accurate data in astronomy: pulsar timing, especially for PSR-1913+16. For massive scalar–tensor gravity, one obtains the upper boundary for the scalar field mass: $m_\varphi < 7 \times 10^{-15}$ (cm$^{-1}$). For hybrid metric-Palatini f(R) gravity this looks as $\phi_0 \leq 0.00004$, $m_\varphi \leq 1.4 \times 10^{-14}$ (cm$^{-1}$).

The next possibility appears at the Solar System scales. For hybrid metric-Palatini f(R) gravity in a case of a light scalar field $m_\varphi r \ll 1$ and using the experimental values of PPN parameters, one constrains $\phi_0$ as $-8 \times 10^{-5} < \phi_0 < 7 \times 10^{-5}$ from the $\gamma^{\text{exp}}$ at the $2\sigma$ confidence level.

The last possibility that we mention here is the idea of gravity constraining using high-energy physics data from a Large Hadron Collider (LHC). Here, it is interesting to use ATLAS data for four–fermion contact interactions at $\sqrt{s} = 8$ TeV and obtain lower limits on the scale on the lepton–lepton–quark–quark contact interaction $\Lambda$ between 15.4 TeV and 26.3 TeV. Therefore, the scale $M_\star$ (a character value of space–time non-locality) appears to be larger than $3 \times 10^{-11}$ GeV.

Of course, this list is far from being complete and only a few examples were demonstrated. As the accurateness of experiments and observations continues to grow, new possibilities could arise in the near future.

**Author Contributions:** The idea of the paper, conceptualization and the text draft except Section 3 were proceed by S.A., Section 3 was proceed by V.P. All authors have read and agreed to the published version of the manuscript.

**Funding:** This research was supported by the Interdisciplinary Scientific and Educational School of Moscow University "Fundamental and Applied Space Research".

**Conflicts of Interest:** The authors declare no conflicts of interest.

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
