# Peer review of "Extended Gravity Constraints at Different Scalesâ€"

_universe, doi:10.3390/universe8050283_

Round 1
Reviewer 1 Report
Reviewer's comments
Manuscript ID: universe-1652316
Type of manuscript: Review
Title: Extended Gravity Constraints at Different Scales
Authors: Stanislav Alexeyev, Vyacheslav Prokopov
Submitted to section: Gravitation
In this review paper the authors discussed few possibilities to constrain extended gravity models at different space-time and energy scales using astrophysical and high energy physics data. They discussed constrain of extended gravity models at Galaxy clusters scales, for black hole shadows, gravitational wave astronomy, binary pulsars, Solar system and Large Hadron Collider.
I believe that this review paper is interesting and useful for the readers in the field of modified gravity models. For the benefit of the readers I would request some minor changes.
Minor requests:
1) Constrains of parameters of different extended gravity models in sub-parsec galactic scales is given in the paper: Universe 2021, 7, 407. It should be mentioned in Introduction.
2) In section 4. Gravitational wave astronomy: deviations from GR, authors discussed “The time delay between gravitational and electromagnetic signals was equal to 1.6 seconds. Such value of time delay puts a strong limitation on graviton mass [5]. Therefore it makes a great cutoff for the models of massive gravity.” It will be very useful for potential readers if authors could provide more extended list of massive gravity models which is in good agreement with the results of LIGO collaboration for graviton mass limitation.
3) Section 8. Conclusions should be improved and extended.
Authors have to put in Conclusion how investigated models matched with astrophysical data (can they pass the astrophysical data test) and what are their findings for constrains of parameters in case of different extended gravity models.
I would like to recommend publication of this paper after minor revision.
Author Response
==> First of all we would like to thank the Referee for the critical notes that we follow to improve the text. Our answers began with ``==>’’.
In this review paper the authors discussed few possibilities to constrain extended gravity models at different space-time and energy scales using astrophysical and high energy physics data. They discussed constrain of extended gravity models at Galaxy clusters scales, for black hole shadows, gravitational wave astronomy, binary pulsars, Solar system and Large Hadron Collider.
I believe that this review paper is interesting and useful for the readers in the field of modified gravity models. For the benefit of the readers I would request some minor changes.
Minor requests:
1) Constrains of parameters of different extended gravity models in sub-parsec galactic scales is given in the paper: Universe 2021, 7, 407. It should be mentioned in Introduction.
==> Thank you for pointing our attention, the reference is added.
2) In section 4. Gravitational wave astronomy: deviations from GR, authors discussed “The time delay between gravitational and electromagnetic signals was equal to 1.6 seconds. Such value of time delay puts a strong limitation on graviton mass [5]. Therefore it makes a great cutoff for the models of massive gravity.” It will be very useful for potential readers if authors could provide more extended list of massive gravity models which is in good agreement with the results of LIGO collaboration for graviton mass limitation.
==> We added few sentences in the end of Sec.4 to extend the list of the viable models according to this note.
3) Section 8. Conclusions should be improved and extended.
Authors have to put in Conclusion how investigated models matched with astrophysical data (can they pass the astrophysical data test) and what are their findings for constrains of parameters in case of different extended gravity models.
==> We extended the conclusions by the main results from each section.
Reviewer 2 Report
This ms is aimed to the search for the extended gravity constraints from different scales. Unfortunately, it contains number of problems.
- The discussion and presentation is rather chaotic. For instance, on p.3 they jump from scalar-tensor theory to specific F(R) model without real explanation that these are two different models.
- In the discussion of BH shadows the authors seem to miss most of literature on the subject (see Capozziello et al).
- GWs study is related with hundred fresh works. The authors wrote half of page on that. It looks like trivial subject for them. If so why they mentioned it ?
- The introduction and motivation are not clear. What is the purpose of this work? Just collect several pieces from earlier author works?
- The authors do not even mention the realistic models of F(R) gravity which unify inflation and DE (see reviews e-Print: 1011.0544, and refs therein). Eventually, this is done to hide the problems of restricted F(R) theory they use as an example.
In summary, major careful revision of this ms is requested. Current version of ms is not suitable for publication because the reader does not benefit much from its reading.
Author Response
We put ``==>'' before our answers
The discussion and presentation is rather chaotic. For instance, on p.3 they jump from scalar-tensor theory to specific F(R) model without real explanation that these are two different models.
==> Thank you greatly for your critical points. From the beginning our aim was not to present a global review on how to constrain all the existing extended gravities. Our aim was more narrow: to demonstrate different possibilities to constrain gravity models. Our idea was to go down along the energy scale taking the models as the examples that looks more understandable. Yes, sometimes we give the preference to our own results as we know them much better.
In the discussion of BH shadows the authors seem to miss most of literature on the subject (see Capozziello et al).
==> We added the the brightest (to our opinion papers). Of course, we are open to add something else that the Referee treats as important for the reader.
GWs study is related with hundred fresh works. The authors wrote half of page on that. It looks like trivial subject for them. If so why they mentioned it ?
==> Note that the subject of how gravitational wave astronomy could constrain extended gravity models is much more wide. We restrict our consideration with GW170817 test as it seems to be most intriguing not to extend the text too much. We wanted to distinguish the situation when the paper gives the single perception. One can extend the discussion, for example, by the notes on the forms of gravitational wave solutions in different models and the number of polarisation modes. We added this note to the text and the end of GW section.
The introduction and motivation are not clear. What is the purpose of this work? Just collect several pieces from earlier author works?
==> We are very sorry that such the impression could appear. The aim was to give the single perception on different methods used to constrain extended gravity. Therefore we restrict the text by most bright examples both in the methods and in the models. Yes sometimes we base on our own activity as we now it much better. The general aim was to create the text that can be useful for last year graduate students in the frames of the course <<Extended theories of gravity>>. This text covers the themes that are not covered by the textbooks. Our idea is that such a text is preparing stage before the serious reviews.
The authors do not even mention the realistic models of F(R) gravity which unify inflation and DE (see reviews e-Print: 1011.0544, and refs therein).
==> We decided not to pay a lot of attention to the famous R + R^2 models treating that this require additional separate paper. The note on the mentioned paper is added.
Eventually, this is done to hide the problems of restricted F(R) theory they use as an example.
In summary, major careful revision of this ms is requested. Current version of ms is not suitable for publication because the reader does not benefit much from its reading.
==> We made a revision mostly in the conclusions part adding the results for each mentioned model. As for the summary we made changes trying not to change the general task of the manuscript that we discussed above.
Reviewer 3 Report
please see the file

Author Response
We would like to thank the Referee for the critical notes that help us to improve the text. Our answers start after ``==>''
In the introduction, only few words are spent to introduce Hornedeski, DHOST, f(R), and TEGR theories. Nevertheless, the reader may not know what necessities lead to investigate alternative theories, and what the latter are.
==> The few sentences on why one has to extend GR with more detailed explanation is added at the first paragraph of the manuscript.
The sections mix astrophysical scale and gravitational regimes. For example: galaxy clusters are on large astrophysical scales but in weak field regime, while for black hole in just the opposite. Maybe, the authors can re-order their sections thinking to a precise ordering.
==> The idea was to go down via distance scale. It occurred because firstly it was a lecture for astronomers and such the logic seems to be more understandable. Yes, we agree, that the energy scale (weak/strong field approximations) could also be taken as the basis but we suppose that the distance scale that we have chosen is also pretty visual.
I believe that two things are totally missing and needed: first, a section summarizing the state of art of current constraint in General Relativity pointing out the need to extend it to other (and more complex) theories; second, an explanation of the theories under consideration (e.g. a section describing f(R), or at least a paragraph). Authors should state on which theories they want to focus and the reasons.
==> We extended the introduction part trying to answer on these Referee’s critical notes.
I have some concerns of the Eq. (4) and (5). It could be fine to take the Schwarzschild metric, but this should be an inner solution and therefore field equation cannot be in the vacuum. Can the authors say something about that?
==> Here the situation is more complicated. The usage of Schwarzschild metric is a consequence of the fact that the matter distribution in the discussed cluster is appropriate. Really, as it is well-known from the literature spherically-symmetric solutions for the extended models must have 2 metric functions. As the Schwarzschild metric could be treated as first 2 terms of the Taylor expansion correspondingly $r^{-1}$ the Schwarzschild metric usage could be treated as first approximation. Formally this analysis must be extended. We stop this procedure because of the observational data errors. We add this statement to the text.
There is a lot of literature missing on galaxy clusters.
==> Thank you for pointing out our attention to these papers we add them to the text.
Reviewer 4 Report
In this article, several possible ways to constrain extended gravity models at Galaxy clusters scales, namely, the regime of dark energy explanations and comparison with $\Lambda$CDM are reviewed. In particular, black hole shadows, gravitational wave astronomy, binary pulsars, Solar system and Large Hadron Collider for high energy physics at TeV scale are argued. The discussions could be interesting and descriptions are presented in detail. Thus, if the following points are reconsidered carefully, this article could be reconsidered for publication.
1. It is recognized that this article is a kind of review article. If it is the case, it would be useful for readers to explain in Abstract that this article is a review. Moreover, in Abstract, a short summary of this review article should be described.
2. There exist a number of the past related review articles on the issue of dark energy and modified gravity theories. By comparing with these preceding review articles, the new and significant features of this work should be stated more explicitly and in more detail. That is, the differences between this review and the past ones should be explained in more detail and more clearly.
Author Response
- It is recognized that this article is a kind of review article. If it is the case, it would be useful for readers to explain in Abstract that this article is a review. Moreover, in Abstract, a short summary of this review article should be described.
==> According to the Referee suggestion we note from the beginning of the Abstract that this is the review. Also we add a phrase to the abstract that the key idea is that modern experimental and observational precise data gives a chance to go beyond the general relativity.
- There exist a number of the past related review articles on the issue of dark energy and modified gravity theories. By comparing with these preceding review articles, the new and significant features of this work should be stated more explicitly and in more detail. That is, the differences between this review and the past ones should be explained in more detail and more clearly.
==> To follow the Referee’s suggestion we rewrite the first paragraph in the Conclusions section to point out the readers attention that we briefly discuss a set in a unified way paying attention on common feathers of all mentioned items, to help one to study carefully each item references to corresponding reviews are provided.
In general we would like to kindly thank the Referee for the ideas of the text improving.
Round 2
Reviewer 2 Report
The authors completely ignored my previous report. No revision is done except of several added sentences in conclusion. In such situation, I only can recommend to authors to really do major revision. In present form, it is not suitable for publication.
Author Response
Dear Reviewer. In the previous round we tried to give our answers on your objections. Really, we were not agree with some of them and I tried to argue my position. I treat the review process as the dialog with an expert but the author also has a right for his point of view. Unfortunately in the second round your reaction appeared to be negative without additional arguments. I am very sorry that we have the opposite points of view and our arguments were not accepted or critically discussed by you. In such a case we continue to insist on our version. Of course, we are ready to the negative decision.
Reviewer 3 Report
I recommend accepting it for publication.
Author Response
Thank you very much!